# Efficacy of Wheelchair Skills Training Program in Enhancing Sitting Balance and Pulmonary Function in Chronic Tetraplegic Patients: A Randomized Controlled Study

**DOI:** 10.3390/medicina59091610

**Published:** 2023-09-06

**Authors:** Seung-Min Nam, Dong-Kyun Koo, Jung-Won Kwon

**Affiliations:** 1Department of Sports Rehabilitation and Exercise Management, Yeungnam University College, Daegu 42415, Republic of Korea; ngd1339@naver.com; 2Department of Public Health Sciences, Graduate School, Dankook University, 119 Dandae-ro, Dongnam-gu, Cheonan-si 31116, Chungcheongnam-do, Republic of Korea; definikk@gmail.com; 3Department of Physical Therapy, College of Health Sciences, Dankook University, 119 Dandae-ro, Dongnam-gu, Cheonan-si 31116, Chungcheongnam-do, Republic of Korea

**Keywords:** cervical spinal cord injury, wheelchair skills training, pulmonary function

## Abstract

*Background and Objectives*: This study aimed to evaluate the effectiveness of a wheelchair skills training program (WSTP) in improving sitting balance and pulmonary function in patients with chronic tetraplegia resulting from cervical spinal cord injury (cSCI). *Materials and Methods*: Twenty-four patients were randomly divided into WSTP and control groups. The WSTP group participated in the WSTP for eight weeks, while the control group underwent conventional physical therapy for the same eight-week period. Sitting balance was evaluated using the activity-based balance level evaluation (ABLE) scale, and pulmonary function was evaluated using forced vital capacity (FVC), forced expiratory volume in one second (FEV1), and peak expiratory flow (PEF). *Results*: The WSTP group showed significant improvements in both sitting balance and pulmonary function during the intervention period (*p* < 0.05), whereas the control group did not show any significant changes. A strong positive correlation was found between ABLE scores and all three pulmonary function parameters across all time points. *Conclusions*: Our results suggest that the WSTP significantly improves sitting balance and specific aspects of lung function in patients with tetraplegia.

## 1. Introduction

Chronic cervical spinal cord injury (cSCI) is a debilitating condition that significantly affects the quality of life of the affected individuals [1]. This condition is characterized by the loss of motor and sensory functions, leading to profound physical limitations [2]. Despite its importance, pulmonary function impairment is often overlooked as a critical consequence of cSCI. The injury disrupts the neural pathways controlling the diaphragm and intercostal muscles, resulting in pulmonary insufficiency, which can further worsen the patient’s condition [3,4].

Several studies have documented the effect of cSCI on the pulmonary function [2,3,4]. Previous studies have demonstrated that patients with cSCI frequently experience a decline in vital capacity, forced expiratory volume (FEV), and maximal inspiratory and expiratory pressures [5,6,7]. These impairments may lead to various complications including sleep-disordered breathing; recurrent pulmonary infections; and in severe cases, pulmonary failure [8,9]. The wheelchair skills training program (WSTP) has emerged as a promising intervention to improve physical functioning among patients with cSCI. The WSTP involves various wheelchair-related tasks, such as wheelchair propulsion, navigating obstacles, and transferring to and from a wheelchair. The primary goal of the WSTP is to enhance mobility, independence, and overall quality of life [10,11].

Previous research has shown that the WSTP enhances wheelchair skills and mobility in individuals with cSCI [12,13]. While the potential impact of the WSTP on pulmonary function remains understudied, it is reasonable to hypothesize that the WSTP could have a positive effect on respiratory health. This is based on the notion that wheelchair propulsion involves significant engagement of the upper body, which could potentially contribute to better pulmonary muscle strength and endurance [14,15,16]. Nevertheless, there is limited conclusive research in this area, underlining the need for more focused investigation. Additionally, it is posited that the WSTP may improve sitting balance through training that emphasizes core stability and postural control.

Consequently, this study investigated the effects of a wheelchair skills training program on sitting balance and pulmonary function in patients with tetraplegia. Furthermore, this study aimed to compare the outcomes of the WSTP group with those of the control group, who received conventional physical therapy. The main purpose of this research was to understand the potential benefits of wheelchair skills training on both balance and pulmonary function in patients with tetraplegia, as well as to provide a comparative analysis with conventional physical therapy methods.

## 2. Materials and Methods

### 2.1. Subjects

A randomized controlled trial was conducted in this study. A total of 24 patients with tetraplegia were included in this study and assigned to either the training (WSTP) or control (conventional physical therapy; G-POWER: F tests; analysis of variance (ANOVA): repeated measures, within–between interaction; two-way, effect size: 0.3, α-err-prob: 0.05, power [1 − β err-prob:0.8] = 20) groups. The training and control groups consisted of 12 patients each. The inclusion criteria for participation were as follows: (1) diagnosed as having a cervical spinal cord injury (C-SCI) and categorized as B/C according to the American spinal injury association impairment scale (ASIA) classification system; (2) use of a manual wheelchair for independent mobility; (3) absence of severe pain or muscle spasm in sitting position that affects body function; (4) no significant visual or vestibular impairment; and (5) no psychiatric or neurological disorder that could affect the study. Participants who did not attend the 8 week training program or exhibited unstable psychological or mental states were excluded from the study. The study protocol was approved by the local ethics committee of the university, and informed consent was obtained from each patient prior to data collection. The study was conducted in accordance with international ethical standards [17].

### 2.2. Measurements

#### 2.2.1. Activity-Based Balance Level Evaluation (ABLE)

The ABLE scale was used to assess the sitting balance of individuals with cSCI. This tool consists of 28 items that evaluate balance while sitting, standing, and walking [18]. For the purpose of this study, seven items specifically related to sitting balance were selected for evaluation. Each item was scored on a five-point scale ranging from 0 to 4, with a score of zero indicating the lowest level of function and a score of 4 indicating the highest level of function (see Appendix A). The evaluations were conducted while the participants were seated in their regular wheelchairs (with a mean height of 45–48 cm) under the guidance and supervision of a physical therapist with over 5 years of experience. No orthosis or additional equipment was used unless necessary to maintain posture. Each participant performed the assessment twice, and the highest score was recorded.

#### 2.2.2. Pulmonary Function

Pulmonary function was measured using a CardioTouch 3000-S spirometer (Bionet Co., Ltd., Seoul, Republic of Korea). The evaluated variables included forced vital capacity (FVC), forced expiratory volume at one second (FEV1), and peak expiratory flow (PEF). The participants were instructed in detail about each test procedure and positioned in a consistent seated posture throughout the duration of the measurements. The participants were instructed to take maximal inspiration, followed by maximal expiration, producing a flow-volume loop. For the FVC test, the participants were asked to inhale maximally and exhale as rapidly and completely as possible after taking three or more breaths as usual [19]. During the FVC test, FEV1 and PEF were measured without additional procedures. A designated rest period was provided between tests, and if a participant experienced fatigue, dizziness, or breathing difficulties, the test was immediately suspended, and sufficient rest was provided before retesting. All assessments were conducted and supervised by the same physical therapist, who refrained from issuing verbal commands that could influence breathing patterns.

#### 2.2.3. Wheelchair Skills Test

In this study, both groups of participants underwent testing and training using the Wheelchair Skills Test (WST), version 4.1. These sessions were supervised by an experienced physical therapist to ensure safety and accuracy. The WST is designed to assess a wheelchair user’s ability to perform various skills needed in everyday life. The procedures for the WST were based on the guidelines outlined in the Wheelchair Skills Program (WSP) Version 4.1 Manual [20]. The WST 4.1 consists of 32 individual skills, divided into three levels: indoor, community, and advanced. Each skill was scored on a scale of 0 to 2, with “pass” indicating a score of 2, “pass with difficulty” indicating a score of 1, and “fail” indicating a score of 0. These scores were then used to calculate the total WST percentage scores. The WST is widely recognized as a valid, reliable, safe, and practical method for assessing the functional skills of manual wheelchair mobility [21,22].

### 2.3. Interventions

To ensure safety and standardization, the participants used standard rear-wheel drive wheelchairs equipped with solid tires and toggle brakes. The wheelchair configuration was adjusted individually for each participant during the training period. All the participants underwent a 10 min of familiarization on two different occasions/days prior to participating in the intervention, in order to adapt to the standardized wheelchair. Both groups underwent an 8 week intervention consisting of three 1 h sessions per week, including warm-up, training programs, and cool-down activities (Figure 1).

The WSTP was conducted by a physical therapist trained in the WSTP and followed version 4.1 of the program sessions [22,23] (see Appendix A). Each skill and training series continued until the skills were mastered or until the physical therapist and participant determined that it was time to progress. Prior to beginning the training program, the participants received information about the training methods and potential injuries that could occur during the exercise.

The control group underwent conventional physical therapy concurrently with the training group. Conventional physical therapy included upper extremity strengthening and endurance exercises on an arm ergometer, as well as aerobic exercise with indoor track cycling. The conventional physical therapy was modified based on the participants’ neurological level and individual wheelchair skills. Conventional physical therapy was performed at 70% maximum heart rate intensity (or a Borg rating of 3–4). After completion of the study procedures, the control group was offered the WSTP as an additional incentive to participate.

#### Statistical Analyses

Data analysis was conducted using the SPSS software (version 25.0, IBM Inc., Chicago, IL, USA). The means and standard deviations were calculated for the WST score and ABLE scale, as well as for each pulmonary function variable (FVC, FEV1, and PEF) at baseline, 4 weeks, and 8 weeks in each group. The Kolmogorov–Smirnov test was used to assess the normality of the dependent variables. A 2 × 3 (two groups: WSTP group, control group; three time points: baseline, 4 weeks, and 8 weeks) two-way repeated measures ANOVA design was used for the analysis. Post hoc pairwise comparisons were conducted using the Bonferroni correction to further explore the data. Separate independent t-tests were used to analyze the changes between the groups at each time point. In addition, statistical analysis was used to assess the correlation between the ABLE scores and measures of pulmonary function using Pearson’s correlation coefficient. The significance level (α) was set to 0.05 for all statistical tests.

## 3. Results

Table 1 presents a summary of the demographic data of the participants. The data indicate no significant differences in sex, age, height, weight, SCI level, ASIA scale score, post-injury period, or wheelchair-dependent period between the WSTP and control groups (*p* > 0.05). In addition, no significant differences were observed between the groups in terms of baseline ABLE scores and FVC, FEV1, or PEF (*p* > 0.05). Table 2 presents the total scores for WST, ABLE, FVC, FEV1, and PEF, as well as the interaction effects. In addition, it presents the results of the post hoc analyses of the differences in changes between the groups. All the variables showed a significant effect over time and significant group × time interaction (*p* < 0.05).

### 3.1. Post Hoc Pairwise Comparisons and between-Group Changes over Time

In this study, we performed post hoc pairwise comparisons using the Bonferroni correction. A significant increase in ABLE scores was observed in the WSTP group across all intervention periods (baseline to 4 weeks, 4 to 8 weeks, and baseline to 8 weeks) (*p* < 0.05). Similarly, WST scores in the WSTP group also showed a significant increase across all intervention periods (*p* < 0.05). In contrast, the control group showed a significant change in WST scores only during the 4 to 8 weeks period (*p* < 0.05) and no significant changes at any other time points. Post hoc pairwise comparisons with Bonferroni correction confirmed these findings for both ABLE and WST scores. However, no significant changes were observed in the ABLE scores in the control group at any intervention period (*p* > 0.05). The WSTP group showed a significant increase in FVC values across all intervention periods (*p* < 0.05). In contrast, the control group did not show any significant changes (*p* > 0.05; Figure 2). The WSTP group showed a significant increase in FEV1 across all intervention periods (*p* < 0.05), whereas the control group showed a significant increase only in the baseline to week 4 period (*p* < 0.05), and not at any other time point (*p* > 0.05; Figure 2). Significant increase in PEF measures were observed in the WSTP group across all intervention periods (*p* < 0.05), unlike the control group, which showed no significant change at any time point (*p* > 0.05; Figure 2).

Additionally, we conducted a post hoc comparison of between-group changes over time using an independent *t*-test. The WSTP group demonstrated significantly higher changes in WST and ABLE scores, as well as pulmonary function (FVC, FEV1, and PEF), compared to the control group in the baseline to 4 weeks and baseline to 8 weeks periods (Table 2).

### 3.2. Correlation between ABLE Scores and Pulmonary Function Parameters

The analyses revealed a significant correlation between the ABLE scores and the pulmonary function parameters (FVC, FEV1, and PEF) (*p* < 0.05). At baseline, we found a strong positive correlation between the ABLE score and FVC (r = 0.87), FEV1 (r = 0.91), and PEF (r = 0.97). This suggests that higher ABLE scores are associated with higher values of the three measures of pulmonary function. After four weeks of training, a strong positive correlation was noted between the ABLE scores and FVC (r = 0.92), FEV1 (r = 0.94), and PEF (r = 0.90). This indicates that the close association between high ABLE scores and high measures of pulmonary function persisted as ABLE training progressed. At 8 weeks after training, there was a strong positive correlation between ABLE scores and FVC (r = 0.93) and PEF (r = 0.95). However, the correlation between the ABLE score and FEV1 (r = 0.55) was relatively weak at this time point (Figure 3). 

## 4. Discussion

This study examined the effects of 4 and 8 week WSTP on sitting balance and lung function in patients with tetraplegia resulting from chronic cSCI. The results suggest that the WSTP not only helps with mobility and independence, but also improves specific aspects of lung function, such as FVC, FEV1, and PEF.

Our findings are consistent with those of prior studies demonstrating the effectiveness of the WSTP in improving mobility and independence in individuals with cSCIs [10,24]. In addition, our study expands the current body of literature by demonstrating that such training may enhance the pulmonary function in this population. The WSTP group showed significant progress in all intervention periods, as evidenced by improved ABLE scores [18]. The WSTP improved the ABLE score for sitting balance in tetraplegic patients, which is critical for daily wheelchair use and maintaining independence [18]. The structured and stepwise format of the WSTP likely improved postural control, increased trunk stability, and strengthened upper-body muscles, all necessary for maintaining sitting balance [25,26,27]. In contrast, the control group that did not receive the WSTP may not have received focused training on muscles vital for maintaining balance in a wheelchair. The conventional physical therapy rendered to the control group included a range of exercises aimed at enhancing overall strength, flexibility, and coordination [28,29,30]. In contrast, the control group that received conventional physical therapy did not show significant improvement in ABLE scores. It is worth noting that the conventional physical therapy provided to the control group may not have been as effective in focusing on the specific muscles vital for maintaining balance in a wheelchair, possibly contributing to the lack of substantial change in ABLE scores.

Moreover, this study extended the scope of evaluation to include WST scores. The data demonstrated that all measured variables, inclusive of WST scores, exhibited significant group × time interactions. These results corroborate the notion that WSTP delivers a comprehensive beneficial impact on the functional wellness of individuals afflicted with cSCI. The WST scores, in particular, serve as an encompassing metric that effectively captures multiple dimensions of functional and physical performance. From a neurological perspective, the efficacy of wheelchair skill training programs in enhancing the wheelchair skill test scores of patients with cSCI can be attributed to a combination of factors. These include improved motor learning through neuroplasticity, where the brain’s ability to reorganize itself by forming new neural connections comes into play [31]. Psychological elements such as increased confidence and reduced anxiety, which are neurologically mediated, further improve motor skills [32]. As a result, the incorporation of WST scores further supports the WSTP’s various therapeutic benefits.

Our data showed that FVC improved significantly after both 4 and 8 week interventions. This suggests that the WSTP increases the ability of the lungs to hold air, likely because the physical demand for moving a wheelchair engages the chest, back, and arm muscles [33,34,35]. In addition, the 8 week WSTP intervention may have further strengthened the diaphragm and muscles between the ribs, which are essential for lung expansion [36]. FEV1 also improved significantly after the WSTP, particularly in the 8 week group. The FEV1 measures the amount of air that can be forcefully exhaled in a second. This is positively affected by strengthening the muscles used in exhalation, a likely benefit of the physically demanding WSTP activities [37]. The 8 week program appeared to offer even better muscle conditioning, resulting in greater improvement in FEV1 [38,39]. Improvements in PEF were observed after both the 4 and 8 week WSTPs. Given that PEF measures the maximum speed of forced exhalation, an improvement in PEF following WSTP suggests stronger respiratory muscles, which improve the ability to exhale air forcefully and quickly [37]. Interestingly, the data suggest that the 8 week program may lead to even greater improvements, possibly because of the extended period of muscle training and adaptation [16,40]. Our study primarily focused on the effects of WSTP on mobility skills. The argument that these enhanced mobility skills could lead to improvements in cardiovascular and pulmonary function is speculative. The premise is that improved mobility may encourage more extensive wheelchair use, increasing physical activity levels and thus potentially affecting cardiovascular and pulmonary health [41,42]. It is a limitation of our study that we did not measure the extent of wheelchair use outside of the training sessions. This makes it difficult to quantify the direct impact of the WSTP on cardiovascular and pulmonary health. Future studies could include tracking mechanisms to measure daily wheelchair use and activity levels. The lack of significant changes in FVC, FEV1, and PEF in the control group may be explained by the nature of their training. Unlike the WSTP, conventional physical therapy may not target the muscles directly involved in respiratory mechanics. Conventional physical therapy in the control group may not have included such intensive and specific muscle training and, therefore, could not effectively enhance the strength and endurance of the respiratory muscles. This lack of targeted, intensive respiratory muscle training would explain why the control group did not experience significant improvement in FVC, FEV1, or PEF. 

Our study revealed a strong positive correlation between ABLE scores and pulmonary function (FVC, FEV1, and PEF). While this suggests that WST may have a beneficial impact on pulmonary function, it is important to note that the correlation does not establish causation. Both outcomes may be independently impacted by WST, and further studies are required to elucidate any direct cause-and-effect relationships [28]. The hypothesis is that the WST program strengthens the chest, back, and arms used during wheelchair maneuvering, which may lead to increased stability of the chest wall and potential improvements in pulmonary functions such as FVC, EVC, and PEF [43,44].

This study had several limitations. First, it was conducted with a limited number of participants. The small sample size may limit the generalizability of these findings to a broader population of patients with tetraplegia. Further research with a more diverse and larger sample is necessary to ensure the generalizability of the findings. Second, this study lacked long-term follow-up data, which restricted our ability to understand the long-term benefits of the WSTP. Third, muscle strength or activity was not assessed. Future studies should include such measures to provide a more comprehensive understanding of the effects of the WSTP on individuals with tetraplegia. Fourth, the improvements in sitting balance observed in the study could partly be attributed to increased efficiency in performing wheelchair-related tasks covered by the WST program. We acknowledge the study’s limitation in not measuring wheelchair use outside of training sessions, making it challenging to establish a direct causal link. To address this gap, we recommend future research that includes tracking mechanisms for daily wheelchair use and further empirical studies to validate these hypotheses. However, the use of the ABLE scale for evaluation suggests that there may also be genuine improvements in sitting balance. Further research is needed to distinguish between the two. Fifth, in this study, the use of standardized wheelchairs for all participants presents specific limitations that should be considered when interpreting the results. The standardized wheelchairs may not account for the diverse physical needs and medical conditions of the participants, thereby potentially affecting their comfort and safety. This lack of individualization introduces a confounding variable that could influence the study outcomes. Moreover, the unfamiliarity with the standardized wheelchair may affect the participants’ performance, compromising the ecological validity and generalizability of the study’s findings. Therefore, while the use of standardized wheelchairs facilitated consistency in data collection, it also imposes constraints that warrant careful consideration.

## 5. Conclusions

In conclusion, the present study suggests potential benefits of WSTP in enhancing sitting balance and pulmonary function in patients with chronic cSCI. While our findings indicate higher ABLE scores and improved pulmonary parameters such as FVC, FEV1, and PEF, it is important to note that these outcomes should not be interpreted as definitive evidence of efficacy, given the possibility of alternative explanations for the observed changes.

## Figures and Tables

**Figure 1 medicina-59-01610-f001:**
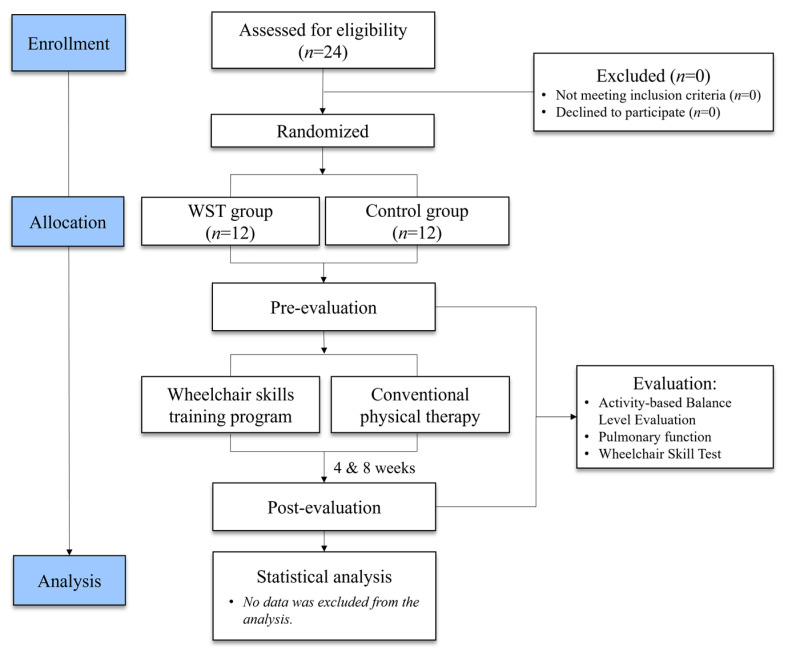
Flowchart of the experimental procedure.

**Figure 2 medicina-59-01610-f002:**
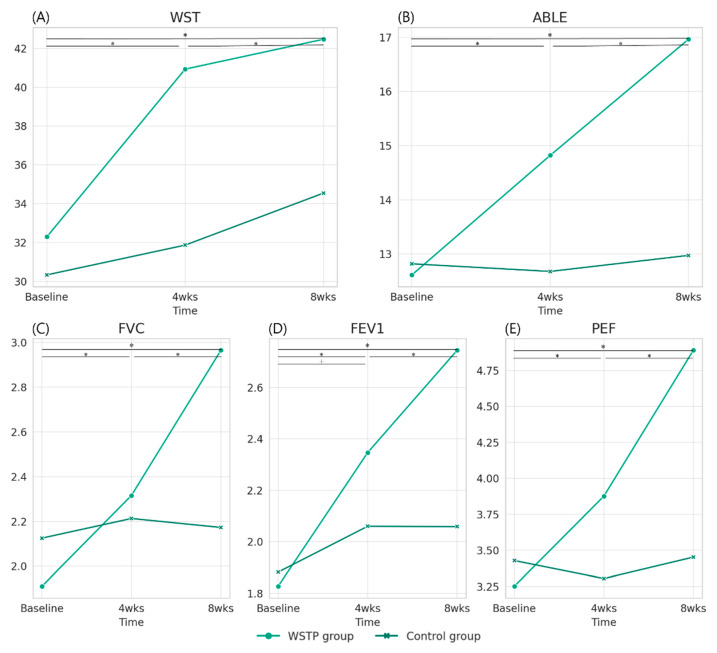
Graphical representation of changes in (**A**) wheelchair skill test scores, (**B**) activity-based balance level evaluation (ABLE) scores, (**C**) forced vital capacity (FVC), (**D**) forced expiratory volume in 1 s (FEV1), and (**E**) peak expiratory flow (PEF) values between the WSTP and control groups over the 4 and 8 week intervention periods. * Significant difference in WST and ABLE scores and FVC, FEV1, and PEF values observed among the time points after the WSTP. ^†^ Significant difference in FEV1 values observed between time points after the conventional physical therapy.

**Figure 3 medicina-59-01610-f003:**
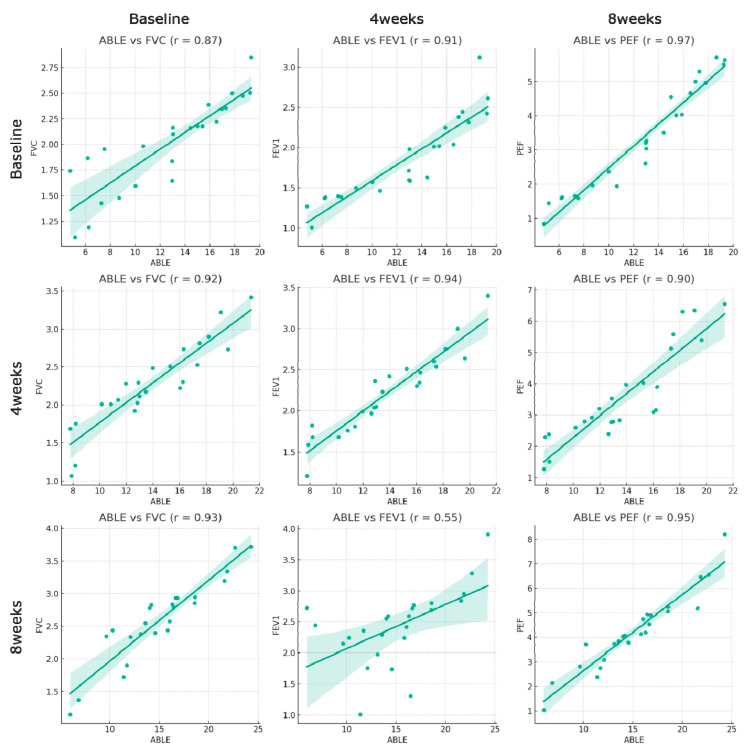
Correlation between ABLE scores and measures of pulmonary function at baseline, and at 4 and 8 weeks after training. The correlation coefficients (r-values) for forced vital capacity (FVC), forced expiratory volume in 1 s (FEV1), and peak expiratory flow (PEF) are shown. A strong positive correlation was observed across all measured data points, indicating a significant relationship between these variables.

**Table 1 medicina-59-01610-t001:** Characteristics of the subjects at the baseline.

	WSTP Group	Control Group
Age (years)	36.02 (7.16)	35.81 (4.51)
Gender (male/female)	9/3	8/4
Height (cm)	167.20 (8.93)	166.50 (7.07)
Weight (kg)	64.52 (7.94)	64.27 (4.69)
SCI level (C5-C6/C7-T1)	7/5	6/6
ASIA scale (B/C)	6/6	5/7
Post-injury period (months)	34.92 (8.35)	36.41 (9.19)
Wheelchair-dependent period (months)	34.22 (8.56)	35.66 (6.58)

Values represent mean (±standard deviation); WSTP: wheelchair skill training program.

**Table 2 medicina-59-01610-t002:** Comparison of WST scores, ABLE scores, and pulmonary function (FVC, FEV1, PEF) across the groups at baseline, 4 weeks, and 8 weeks of the intervention period.

Variables	Group	Baseline	4 Weeks	8 Weeks	F(*p*)			Post Hoc Analysis (Mean Difference)	
					Group	Time	Interaction (Group × Time)	Baseline to 4 Weeks	Baseline to 8 Weeks
WST (score)	WSTP	32.30 (6.84)	40.93 (8.07)	42.47 (7.73)	5.41 * (0.03)	12.56 * (<0.01)	3.51 * (0.04)	W > C	W > C
	Control	30.34 (6.33)	31.87 (6.41)	34.56 (9.04)					
ABLE (score)	WSTP	12.62 (1.34)	14.82 (1.07)	16.96 (1.21)	1.27 (0.27)	52.57 * (<0.01)	41.78 * (<0.01)	W > C	W > C
	Control	12.82 (1.46)	12.68 (1.16)	12.98 (1.31)					
FVC (L)	WSTP	1.91 (0.12)	2.32 (0.16)	2.97 (0.14)	1.34 (0.26)	52.21 * (<0.01)	38.71 * (<0.01)	W > C	W > C
	Control	2.13 (0.13)	2.21 (0.17)	2.17 (0.15)					
FEV1 (L)	WSTP	1.83 (0.15)	2.35 (0.13)	2.75 (0.14)	2.90 (0.10)	41.02 * (<0.01)	11.79 * (<0.01)	W > C	W > C
	Control	1.88 (0.16)	2.06 (0.15)	2.06 (0.15)					
PEF (L)	WSTP	3.25 (0.44)	3.88 (0.42)	4.89 (0.38)	1.03 (0.32)	33.87 * (<0.01)	29.71 * (<0.01)	W > C	W > C
	Control	3.43 (0.48)	3.31 (0.46)	3.45 (0.42)					

Values are presented as mean (standard deviation). Post hoc analysis detailed significant differences in measurements between groups at the baseline to 4 weeks and baseline to 8 weeks intervals. WST: wheelchair skill test, ABLE: activity-based balance level evaluation, FVC: forced vital capacity, FEV1: forced expired volume after 1 s, PEF: peak expiratory flow, WSTP(W): wheelchair skill training program, C: control group; F(p) values are shown for Group, Time, and Interaction (Group × Time); * *p* < 0.05.

## Data Availability

The data cannot be publicly disclosed due to patient privacy protection.

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
