# Peer review of "Efficacy of Wheelchair Skills Training Program in Enhancing Sitting Balance and Pulmonary Function in Chronic Tetraplegic Patients: A Randomized Controlled Study"

_medicina, 2023, doi:10.3390/medicina59091610_

Round 1

Reviewer 1 Report

Dear Authors,

I have read and evaluated your research in detail. It is valuable in its research subject and scope and is almost perfectly written in its current form. I just want you to highlight the main hypothesis after the purpose statement in the introduction section.

Kind regards

Reviewer 2 Report

This is an interesting study that explores the potential of a wheelchair skills training program to produce supplementary benefits to increasing wheelchair mobility skill capacity or performance. While it is intriguing to report a clinical effect on sitting balance and pulmonary function, there are two main issues that are a concern: the authors have not provided a fulsome rationale for why or how skills training would or could have such an effect, and there is insufficient description of the intervention itself which further clouds the cause-effect relationship.

1.     The authors refer, throughout the manuscript, to WST as “wheelchair skills training” making specific reference (line 116) to v4.1 of the Dalhousie University program. This is confusing as (according to the website and literature) WST refers to the “Wheelchair Skils Test”, which is a specific outcome measure of skill capacity/performance (and which incidentally is not reported on in this study). In v4.1, WSTP refers to the Wheelchair Skills Training Program (guidelines on delivery of wheelchair skills training) and collectively the WSTP and WST comprise the Wheelchair Skills Program (WSP). The authors need to be clear about what specifically they were using in this study. Also, you may want to cite the actual training program materials from the https://wheelchairskillsprogram.ca  site.

2.     There is little description of the intervention. Lines 108+ imply that the wheelchair used was not the participant’s own – please clarify if this was the case and if so, why they didn’t use their own (and if this applied to both the treatment and control group). Line 110 – what “training” did participants receive prior to beginning the intervention … was this the WSTP or something else, and was this applicable to both groups?
The description of the treatment group intervention needs considerable elaboration. While the authors describe the “WST” as “structured and stepwise” (line 227), my understanding is that the program is customized to the individual based on their current skill set and goals they set related to skill acquisition (as well as the learning/training activities and sequences used); was the intervention in this study completed in a pre-set sequence and did participants engage in training of all 32 skills in the order listed on the test? Was the training approach/intervention exactly the same for all 12 participants?

3.     Given that the training program intends to address specific skills (as outlined on the test), it would be helpful to know what the participants’ skill proficiency was prior to and after the intervention. Did the authors collect WST scores prior to and after intervention? Since the WSP appears to focus on precision, proficiency and safety of skill performance (i.e., high cardiorespiratory demand is not encouraged and accuracy is prioritized over speed), knowing how many or which skills the participants were not proficient with at the outset would be interesting, as well as whether their proficiency improved as a result of the program. For example, if participants were highly skilled at the outset (which might very well be the case as they had a mean of 3 years of wheelchair use experience), it would be important to know what the training sessions were composed of (i.e., practicing skills that they already could do well?).
Despite reporting a statistically significant improvement in pulmonary function post intervention, the authors have not identified a reasonable rationale or causative connection between treatment and outcome. The background section (lines 47-53) is very brief. Further, it suggests that propulsion requires “substantial strength and endurance”, but the WSP focus is on developing strategies and skills (rather than increasing strength and endurance) and, in fact, teaching techniques to reduce the strength and endurance required (e.g., semi-circular push patterns that incorporate coasting rather than continuous pushing) – these points seem in opposition. If the authors’ intent is suggesting that skills training involves having the participants propel more (than they typically do), then this seems like an important outcome to incorporate in the study – is there any evidence for/against the treatment group simply increasing the amount of time they spent propelling their wheelchair (compared with the control group), and if so, is it skills training or simply more wheelchair use that produces the results (an implication in the Discussion, where it is suggested the extra time spent wheeling was responsible for improvement, rather than “skill training” – p 239)? There appears to be a premise that skills training has a cardiovascular focus when, in fact, it is about learning mobility skills.
The 3 citations in the background (14-16 on p. 2)
do not support the sentence that precedes (relationship between wheelchair skills training and cardiorespiratory/pulmonary benefit); El-Kader used resistive exercise, not wheelchair propulsion or skills training; neither Desroches nor Chow propose the relationship and neither refer to pulmonary function at all, nor strength or muscular benefits of propulsion (in fact argues MWC propulsion can have long-term negative physiological outcomes).
Furthermore, there is no rationale presented for why or how wheelchair skills training might improve sitting balance. This needs to be addressed in the background section. Without some foundational basis or theory for cause and effect, the findings of the study might be simply a spurious or confounding result.

4.     In the Discussion section:
Line 220-21 I don’t think this is accurate; you didn’t measure mobility and independence so the current study can’t be compared to previous studies on this basis (unless you did score your participants on the WST …).
Line 233 – you suggest the conventional therapy is beneficial but it appears there was no benefit at all among the control group participants (over time); is this a concern with respect to providing conventional therapy?

Line 262 – it is not clear to me how the association (i.e., correlation not causation) between balance and pulmonary function “provide convincing evidence” of skills training improving PF. These two outcomes are discrete and different; apparently skills training impacts both in this study, but the change in each is independent.

5.     In reviewing the sitting balance subscale, there appear to be a number of items that relate specifically to skills addressed in the WST (e.g., reaching, weight-shift, transfers); is there a possibility that the changes identified reflect being able to perform the skill more efficiently (versus an actual improvement in sitting balance)?

6.     The conclusion of “convincing evidence of efficacy” is overstated; I would tempter this conclusion. There appear to be some alternative reasons for the observed changes.

Reviewer 3 Report

I appreciate the opportunity to review your manuscript. WST is important in people with SCI and the effects of WST training on several outcomes are needed. Though this study is not novel, it confirms the relationships between WST, sitting balance, and pulmonary function.

Abstract

For how long was the control group under physical therapy?

Introduction

Please correct: There is a clear need for more thorough

Methods

Inclusion criteria: complete or incomplete injury? Which ASIA impairment Scale was included?

The effect size used for the sample size calculation is small. Maybe because authors wanted to justify their small sample size. The sample size calculation does not seem trustful.

A more valid and reliable sitting balance measure that is specific for non-ambulatory individuals with SCI could have been used. ABLE has items of standing and walking which are not necessarily suitable for cSCI.

How many years of experience had the physical therapist?

Which post-hoc test was used? Need to specify in statistical analysis

Lines 195-196: not sure what this sentence is here for.

As highlighted above a typo was noted.

Round 2

Reviewer 2 Report

Thank you for sharing your revised manuscript. Some of the previous concerns have now been addressed; however, several concerns are outstanding. In the revision, there is really no expansion on the background, including a rationale or justification for the intervention, beyond modifying the wording to speculate that wheelchair propulsion could increase strength and endurance (and continue to use the same citations which do not support this notion – see previous review).  

2.3 Intervention

The revision of “a 10-minute familiarization process for two days” is still confusing – does this mean 10 min of familiarization on two different occasions/days or 10 minutes across two days?

There has been no elaboration on the details of the intervention in this revision, identified in the first review.

I do not concur with the author’s argument regarding use of “standardized” wheelchair among all participants – wheelchair prescription is a complex process that requires customized device selection and configuration to promote optimal use and function; requiring all participants to use a chair that is not their own (and regularly used) does not promote optimum function, capacity for skill acquisition, nor safety in use – in fact, I would argue the opposite is true. The arguments around maintenance and purchase cost may be promoted with respect to a health system level wheelchair provision service but is not a rationale for the scientific integrity of the study conducted. The decision to use a standard wheelchair for the study cannot be changed, so I recommend identifying the reasons for, and limitations of, this decision rather than justification for this choice.

The position that “skills training indeed focuses on learning mobility skills, but we argue that these skills, by enabling more efficient and extended wheelchair use, also contribute to potential improvements in cardiovascular health and pulmonary function” presents some challenges. Was the benefit from the WSTP training sessions or did the participants use their wheelchair more extensively during the rest of their week (because of the skills acquired)? This will be difficult to quantify since wheelchair use was not measured in the study, but the argument needs to be articulated (i.e., causal chain). I still struggle with the author’s position that participants would extend their wheelchair use to the point of changing cardiovascular and pulmonary function, given that current evidence suggests even experienced/proficient manual wheelchair users predominantly engage in small bouts of activity (~20 seconds in duration) at slow speeds (~ 0.5 m/s) with many stops and turns (see Sonenblum, Sprigle & Lopez 2012 for example) rather than “extended wheelchair use”. Regardless of whether we agree on this point, a fulsome explanation of the apparent benefit is in order.

You’ve included the WST scores – I assume (from the description provided in the Measurements section, that these are % scores out of 100. The mean post-treatment scores of 35 and 42% (and lower pre-treatment scores) seem very different from other published studies where community-dwelling wheelchair users typically enter the WSTP with at least 65% and finish the program with at least 75% capacity scores (see Keeler et al 2018); your study values are quite low. Do you have any explanation for this?
